# Multi-Strain Probiotic Regulates the Intestinal Mucosal Immunity and Enhances the Protection of Piglets Against Porcine Epidemic Diarrhea Virus Challenge

**DOI:** 10.3390/microorganisms13081738

**Published:** 2025-07-25

**Authors:** Xueying Wang, Qi Zhang, Weijian Wang, Xiaona Wang, Baifen Song, Jiaxuan Li, Wen Cui, Yanping Jiang, Weichun Xie, Lijie Tang

**Affiliations:** 1College of Veterinary Medicine, Northeast Agricultural University, Harbin 150030, China; tycoon28644@163.com (X.W.); a502694221@foxmail.com (Q.Z.); wangweijian0305@163.com (W.W.); xiaonawang@neau.edu.cn (X.W.); lijiaxuan9301@gmail.com (J.L.); cuiwen@neau.edu.cn (W.C.); jiangyanping@neau.edu.cn (Y.J.); 2National Key Laboratory of Veterinary Public Health and Safety, College of Veterinary Medicine, China Agricultural University, Beijing 100193, China; songbaifen0819@163.com; 3Key Laboratory of Dairy Science, Ministry of Education, Department of Food Science, Northeast Agricultural University, Harbin 150030, China

**Keywords:** porcine epidemic diarrhea virus (PEDV), multi-strain probiotic, antimicrobial activity, antiviral activity

## Abstract

Porcine epidemic diarrhea virus (PEDV) infection induces severe, often fatal, watery diarrhea and vomiting in neonatal piglets, characterized by profound dehydration, villus atrophy, and catastrophic mortality rates approaching 100% in unprotected herds. This study developed a composite probiotic from Min-pig-derived *Lactobacillus crispatus* LCM233, *Ligilactobacillus salivarius* LSM231, and *Lactiplantibacillus plantarum* LPM239, which exhibited synergistic growth, potent acid/bile salt tolerance, and broad-spectrum antimicrobial activity against pathogens. In vitro, the probiotic combination disrupted pathogen ultrastructure and inhibited PEDV replication in IPI-2I cells. In vivo, PEDV-infected piglets administered with the multi-strain probiotic exhibited decreased viral loads in anal and nasal swabs, as well as in intestinal tissues. This intervention was associated with the alleviation of diarrhea symptoms and improved weight gain. Furthermore, the multi-strain probiotic facilitated the repair of intestinal villi and tight junctions, increased the number of goblet cells, downregulated pro-inflammatory cytokines, enhanced the expression of barrier proteins, and upregulated antiviral interferon-stimulated genes. These findings demonstrate that the multi-strain probiotic mitigates PEDV-induced damage by restoring intestinal barrier homeostasis and modulating immune responses, providing a novel strategy for controlling PEDV infections.

## 1. Introduction

Porcine epidemic diarrhea (PED) is a highly contagious viral disease in pigs, marked by severe diarrhea, vomiting, and dehydration, especially in newborn piglets, leading to high mortality and economic losses in the swine industry [1,2]. The disease is caused by the porcine epidemic diarrhea virus (PEDV), part of the Alphacoronavirus genus, and mainly spreads through the fecal-oral route, with possible airborne transmission [3,4]. The impact of cross-species PEDV spillover in the maintenance and transmission of PEDV requires additional verification [1]. PEDV can infect pigs of all ages, and, compared with nursing pigs, weaned pigs have a shorter recovery from PED [1]. The economic repercussions of PEDV on the swine industry are considerable, necessitating the investigation of diverse preventive and therapeutic strategies [5,6]. A promising avenue of research involves the application of probiotics, such as Lactobacillus species, which are recognized for their advantageous effects on gut health and immune modulation [7,8,9]. For instance, *Lactobacillus crispatus* has been shown to enhance intestinal barrier integrity, sustain intestinal mucosal barrier homeostasis, and mitigate the intestinal inflammatory response [10].

In experimental contexts, research has demonstrated that piglets supplemented with *Lactobacillus rhamnosus* GG show enhanced survival rates, decreased viral loads, and mitigated clinical symptoms and intestinal damage following infection with PEDV [11]. Furthermore, the antiviral efficacy of metabolites from *Lactiplantibacillus plantarum*, particularly exopolysaccharides, is attributed to their ability to prevent PEDV adsorption, reduce inflammatory responses, and induce early apoptosis in infected cells [12]. Additionally, the screening of lactic acid bacteria strains has identified *Lactobacillus agilis* and *Ligilactobacillus salivarius*, isolated from the feces of nursing piglets, as exhibiting significant antiviral activity against PEDV [13]. This activity is mediated through a reduction in pro-inflammatory cytokine expression and the enhancement of cell viability in infected cells [13]. These findings underscore the potential of these *Lactobacillus* strains as natural antiviral agents for the protection of pigs against PEDV infections.

The concept of strain synergy is crucial in the development and functionality of complex probiotics, primarily facilitated by metabolic interactions among different strains, which can result in enhanced probiotic effects [14]. For example, the co-cultivation of *LimosiLactobacillus reuteri* ZJ625 and *Ligilactobacillus salivarius* ZJ614 has been shown to significantly alter their metabolite profiles, thereby affecting metabolic and biosynthetic pathways [15]. Moreover, the combination of various probiotic strains can elicit a more robust immune response. Research on a probiotic consortium derived from poultry strains has demonstrated that the combination of *Lactobacillus brevis* and *Ligilactobacillus salivarius* strains significantly improved growth performance and modulated immune responses in broiler chickens, highlighting the potential of multi-strain probiotics in enhancing host health [16]. Metabolic interactions between strains can boost health by improving intestinal barrier function and reducing inflammation. A study showed that a multi-species probiotic mix effectively prevents leaky gut by maintaining epithelial integrity and lowering inflammation in human intestinal cells, emphasizing the significance of strain synergy for gut health [17].

The distinctive gut microbiota of Min pigs may significantly contribute to resistance to pathogenic infections, as demonstrated by recent studies investigating the interactions between gut microbiota and host immunity [18]. In this research, a composite probiotic was formulated, comprising *Ligilactobacillus salivarius*, *Lactobacillus crispatus*, and *Lactiplantibacillus plantarum*, which were isolated from the intestines of Min pigs. The protective effects of this composite probiotic were assessed on IPI-2I cells infected with PEDV. Concurrently, an in vivo experiment was conducted on piglets to evaluate the probiotic’s impact on pathological changes, inflammatory factors, intestinal barrier integrity, and antiviral signaling pathways in piglets infected with PEDV. This investigation aims to elucidate the protective effects of the composite probiotic on piglets and offers a novel approach for the prevention of PED and the development of microecological live bacterial preparations to mitigate intestinal damage in piglets caused by PEDV infection.

## 2. Materials and Methods

### 2.1. Cells, Bacteria, and Virus

*Staphylococcus aureus* CMCC26003 (*S. aureus* CMCC26003), *Salmonella* Typhimurium SL1344 (*S.* Typhimurium), *Serratia marcescens* ATCC8100 (*S. marcescens* ATCC8100), *enterotoxigenic Escherichia coli* K88 (ETEC K88), and porcine epidemic diarrhea virus HEB strain (PEDV-HEB) were maintained by the Laboratory of Microbiology and Immunology, College of Veterinary Medicine, Northeast Agricultural University. The IPI-2I cells were commercially acquired from BNCC Biological Technology Co., Ltd. (Beijing, China). In previous research conducted in our laboratory, multiple strains of lactic acid bacteria were isolated from the colon contents of Min pigs with antiviral properties. Three lactic acid bacterial strains exhibiting anti-PEDV activity were preliminarily selected for further characterization: *Lactobacillus crispatus* LCM233 (*L. crispatus* LCM233), *Ligilactobacillus salivarius* LSM231 (*L. salivarius* LSM231), and *Lactiplantibacillus plantarum* LPM239 (*L. plantarum* LPM239).

### 2.2. Growth Curves and Acid Production Analysis

Three lactic acid bacterial strains (*L. salivarius* LSM231, *L. crispatus* LCM233, and *L. plantarum* LPM239), isolated from the colonic contents of Min pigs, were selected for this study. The three lactic acid bacterial strains were activated to the third generation and inoculated (1:100 ratio) in antibiotic-free MRS broth. OD_600_ and pH were measured every 2 h from 0 h onward. Concurrently, 1 mL cultures were serially diluted (10^−6^), plated (200 μL on MRS agar), and incubated for viable count determination using the following: CFU/mL = colony count × dilution factor × 5. Growth curves and acid production trends were plotted.

### 2.3. Validation of Synergistic Effects in Multi-Strain Probiotic

Third-generation activated lactic acid bacterial strains (OD_600_ ≈ 1.0) were inoculated in MRS broth as control (single-strain, inoculation ratio of 1:100) and experimental (triple-strain consortium 1:1:1 at inoculation ratio of 1:100) strains. OD_600_ was monitored at 2 h intervals over 24 h by spectrophotometry to test bacterial growth. Comparative growth curves of single-strain and co-cultures were generated to quantitatively assess synergistic growth effects. This foundational analysis established the basis for subsequent investigations of pathogen growth inhibition and antiviral activity in bacterial co-culture systems.

### 2.4. Acid and Bile Salt Tolerance Assay

Third-generation streaking-activated lactic acid bacteria (OD_600_ ≈ 1.0) were inoculated (1:100 ratio) into pH-adjusted MRS broth (5.7 and gradient 3.0–8.0) and MRS broth supplemented with 0.01–0.1% porcine bile salts. After 3 h of incubation (37 °C), the cultures were serially diluted (10^−6^), plated (200 μL on MRS agar), incubated for 24 h (37 °C), and enumerated. The survival rate was calculated as:Survival (%) = (Viable count_test_/Viable count_control_) × 100%

### 2.5. Simulated Gastrointestinal Fluid Tolerance Assay

Preparation of Simulated Gastric Fluid: To prepare the solution, 10 g/L of pepsin was dissolved in 16.4 mL of sterile 0.1 mol/L hydrochloric acid. Subsequently, the solution was filter-sterilized using a 0.22 μm microporous membrane to eliminate macromolecular impurities and microorganisms. Finally, the pH was adjusted to 4.0 using sterile 1 mol/L sodium hydroxide to replicate the gastric conditions of neonatal pigs.

Preparation of Simulated Intestinal Fluid: To prepare the simulated intestinal fluid, 10 g/L of trypsin and 6.8 g of KH_2_PO_4_ were dissolved in 500 mL of deionized water. The pH was adjusted to 6.5 using a 1 mol/L NaOH solution. Subsequently, the solution was sterilized by passing it through a 0.22 μm membrane filter.

Third-generation streaking-activated lactic acid bacteria (OD_600_ ≈ 1.0) were centrifuged (3000 rpm, 4 °C, 5 min), PBS-washed, and resuspended. For treatment, 1 mL bacterial suspension was mixed with 9 mL simulated gastric fluid or 9 mL simulated intestinal fluid. The control contained 1 mL suspension + 9 mL PBS. All samples were incubated (37 °C, 3 h), serially diluted, plated (200 μL on MRS agar), and cultured for 24 h (37 °C). The survival rate was calculated as:Survival (%) = (CFU_treatment_/CFU_control_) × 100%

### 2.6. Antimicrobial Activity Assay

The antibacterial activity of lactic acid bacteria strains and multi-strain probiotic was assessed against *S. aureus* CMCC26003, *S.* Typhimurium SL1344, *S. marcescens* ATCC8100, and ETEC K88 using agar well diffusion. Three lactic acid bacterial strains (OD_600_ ≈ 1.0) were blended at a 1:1:1 ratio and co-inoculated into MRS broth at a final 1:100 dilution. Following stationary-phase cultivation, the cultures were centrifuged (3000 rpm, 4 °C, 5 min), and the supernatants were 0.22 μm filtered to obtain cell-free supernatants (CFS) for subsequent use. CFS were prepared by culturing three lactic acid bacterial strains in MRS broth (1:100 inoculation) to a stationary phase, followed by centrifugation (4 °C, 3000 rpm, 5 min) and 0.22 μm filtration. Concurrently, four indicator pathogens were cultured overnight, adjusted to ≈10^7^ CFU/mL, and mixed 1:100 with molten LB agar. The mixture was poured into plates containing four sterilized Oxford cups. The cups were removed post-solidification. Subsequently, 200 μL CFS (test) or pH 5.7 MRS (control) was added to each well. The plates underwent 4 h of pre-diffusion (4 °C) and 16 h of static incubation (37 °C). The inhibition zone diameters were measured in triplicate.

### 2.7. Electron Microscopic Examination of Lactic-Acid-Bacteria–Pathogen Interactions

Pathogen suspensions (*S. aureus* CMCC26003, *S.* Typhimurium SL1344, and *S. marcescens* ATCC8100, ETEC K88; 10^5^ CFU/mL) were treated 1:1 with lactic acid bacterial supernatant (PBS control) and incubated for 12 h (37 °C). The cells underwent PBS washing, 2.5% glutaraldehyde fixation (>1.5 h, 4 °C), phosphate buffer rinsing (0.1 M, pH 7.2), and graded ethanol dehydration (50%/70%/90% once, 100% twice; 10–15 min/step). The samples were transitioned through tert-butanol series, frozen (−20 °C, 30 min), freeze-dried (HITACHI ES-2030, 4 h), and sputter-coated with 100–150 Å gold (HITACHI E-1010) prior to SEM imaging.

The bacterial samples underwent PBS washing and 2.5% glutaraldehyde fixation (>2 h, 4 °C), followed by centrifugation (3000 rpm, 15 min), 0.1 M phosphate buffer rinses (3 × 15 min), and 1% osmium tetroxide post-fixation (0.5–2 h). After additional buffer rinses (3 × 15 min), the samples were dehydrated through graded ethanol (50%/70%/90% once, 100% twice; 8–10 min/step) and acetone transition. Infiltration occurred via graded acetone, with resin mixtures (1:1, 1:2, 1:3; 0.5 h, 2 h, overnight), followed by embedding, polymerization, and trimming. Ultrathin sections (70–90 nm) were double-stained with uranyl acetate/lead citrate and imaged by transmission electron microscopy (Hitachi H-7650, Tokyo, Japan).

### 2.8. In Vitro Bactericidal Activity Assay

Indicator pathogens (*S. aureus* CMCC26003, *S.* Typhimurium SL1344, and *S. marcescens* ATCC8100, ETEC K88; 1 × 10^6^ CFU/mL) were co-cultured 1:1 with lactic acid bacterial supernatants. At 6, 12, 18, and 24 h, the samples were serially diluted in sterile saline, plated on LB agar, and incubated for 24 h (37 °C). Bactericidal inhibition rates were calculated as: [1 − (CFU_sample_/CFU_control_)] × 100%

### 2.9. Anti-Proliferative Effect of Multi-Strain Probiotic Against PEDV

IPI-2I cells (80% confluency) were treated with maximum non-toxic concentrations of bacterial suspensions or CFS mixed 1:1 with PEDV (100 TCID_50_; 25 μg/mL trypsin). After pre-incubation (37 °C, 2 h), inocula were added to cells (37 °C, 1 h). The viral load was quantified via RT-qPCR 24 h post-infection to assess multi-strain probiotic combination efficacy against PEDV proliferation.

### 2.10. Animals and Experimental Design

A total of 9 Duroc × [Landrace × Yorkshire] pigs (DLY) were used in this study. The animals had unrestricted access to food and water. The DLY pigs were obtained from Gushi Agriculture & Animal Husbandry Group Co., Ltd., Harbin, Heilongjiang Province, China. Following three days of breastfeeding, the piglets were transitioned to formula feeding. All piglets tested negative for PEDV. This study maintained strict environmental controls, setting the temperature at 30 ± 2 °C and relative humidity between 65 and 70%. The piglets were housed individually to eliminate group effects. All experimental procedures for pigs were approved by the Animal Care and Use Committee of Northeast Agricultural University (approval number: NEAUEC20220317). During this study, all animal care and treatment methods were in accordance with the Laboratory Animal Management Regulations (revised 2016) of Heilongjiang Province, China. The animal experimental procedure is described below.

A total of 9 healthy neonatal DLY piglets with similar body weights were randomly allocated into three groups: (1) control group (CON); (2) PEDV-challenged group (PEDV); (3) protected group (complex probiotics + PEDV). The protected group received oral gavage of multi-strain probiotic combination (10^9^ CFU/mL) from postnatal day 1 for three consecutive days, while the CON and PEDV groups received equivalent volumes of saline. On day 4, the PEDV and protected groups were orally inoculated with 1.5 × 10^6^ TCID_50_ of PEDV-HEB strain, and the CON group received equal volumes of DMEM. All piglets were artificially fed identical milk formula. Daily monitoring included clinical observations and body weight measurements. Following euthanasia at the trial endpoint, gross pathological examinations were performed, and jejunum and ileum segments were collected.

### 2.11. Detection Indicators and Methods

#### 2.11.1. RNA Extraction and Quantitative PCR Analysis

Total RNA was extracted using an RNA Purification Kit (Sevenbio, Beijing, China), and cDNA was synthesized with Superscript III reverse transcriptase (Invitrogen, Carlsbad, CA, USA). qPCR was performed in triplicate with LightCycler^®^ 480 SYBR Green I Master on Roche LightCycler 96 under standard cycling: 95 °C/10 s, 60 °C/20 s, and 72 °C/20 s (40 cycles). Melt curve analysis ensured primer specificity. β-actin served as the reference gene. Excel was used for data analysis. Primer sequences are shown in Table 1. Lesion-targeted intestinal specimens were aseptically collected during necropsy, cryopreserved at −80 °C, and reserved for RNA extraction and RT-qPCR quantification of PEDV viral load.

#### 2.11.2. Histopathological Analysis

Vital jejunum and ileum tissues were fixed in 4% buffered formalin for 72 h, embedded in paraffin, and sectioned into 3 µm thick slices. Following dewaxing with ethanol and xylene, sections were stained with hematoxylin–eosin (H&E) and digitally scanned using a Pannoramic Scanner (CaseViewer 2.4, 3DHISTECH, Budapest, Hungary).

#### 2.11.3. AB-PAS Staining of the Small Intestine

Paraffin sections were dewaxed to water through xylene I (20 min), xylene II (20 min), absolute ethanol I (5 min), absolute ethanol II (5 min), and 75% ethanol (5 min), followed by distilled water rinsing. Sections were stained with Alcian blue for 5 min and washed in distilled water (2 min), treated with periodic acid solution (15 min), and rinsed twice in distilled water, then incubated in Schiff reagent for 30 min (light-protected) and flushed under running water (5 min). Hematoxylin counterstaining (3–5 min) was performed, followed by distilled water washing, acid differentiation, hydrochloric-acid–ethanol differentiation, and 3-min bluing in running water. Sections were dehydrated through graded ethanol, cleared in xylene, and mounted with neutral gum. Stained tissues were examined microscopically for image acquisition and analysis.

#### 2.11.4. Electron Microscopic Examination of Small Intestinal Tissues

For ultrastructural analysis, samples were prepared for both scanning and transmission electron microscopy. For SEM, tissues were sectioned into ~2 mm × 5 mm pieces, fixed in glutaraldehyde, dehydrated in ethanol, critically point-dried, sputter-coated with gold (100–150 Å; Quorum, Lewes, UK), and imaged using a JEOL SEM (Tokyo, Japan).

For TEM, ~1 mm^3^ pieces were fixed in 2.5% glutaraldehyde, washed in PBS, post-fixed in 1% osmium acid, dehydrated, embedded, trimmed, and ultrathin-sectioned. Sections were double-stained with uranyl acetate and lead citrate, then examined using a Hitachi H-7650 TEM (Tokyo, Japan) to assess organelle morphology in the jejunum and ileum.

### 2.12. Statistical Analysis

Data are expressed as mean ± standard deviation (SD). Statistical analyses were performed using GraphPad Prism software (version 10.1.2), employing one-way ANOVA and *t*-tests. Significance levels were denoted as follows: * *p* < 0.05, ** *p* < 0.01, *** *p* < 0.001, with “ns” indicating non-significant results.

## 3. Results

### 3.1. Biological Characterization of Probiotics Isolated from Min Pigs

Growth curve analysis defined phases 0–4 h (lag phase), 4–14 h (logarithmic phase), 14–20 h (stationary phase), and post-20 h (decline phase). During logarithmic growth, acid production increased linearly, indicating peak metabolic activity. All lactic acid bacterial strains reduced the medium pH from 5.7 to 4.0–4.5 within 24 h. The viable counts rose until 14 h, then declined, establishing 14 h as the optimal culture duration for subsequent experiments (Figure 1A). This study demonstrated that all three strains of lactic acid bacteria were capable of surviving in both acidic (pH 3.0) and alkaline (pH 8.0) environments (Figure 1B). Furthermore, the results from the bile salt tolerance assay revealed that, in the MRS medium supplemented with 0.1% porcine bile salt, all three strains exhibited survival capabilities, with LCM239 displaying superior tolerance to bile salts compared to the other strains (Figure 1C). In vitro, the gastrointestinal survival assays demonstrated differential tolerance among the three probiotic strains following 3 h of exposure to simulated gastric and intestinal fluids. As delineated in Table 2, LPM233 exhibited the highest gastric fluid survival rate (81.82 ± 12.86%), whereas LSM231 showed maximal resistance to intestinal fluid stress (58.4 ± 7.92%).

### 3.2. Antibacterial Activity of LSM231, LCM233, and LPM239

Strain LCM233 demonstrated maximal inhibition (61.69%) against ETEC K88 (Figure 2A), while LSM231 showed peak efficacy (83.84%) versus *S. aureus* CMCC26003 (Figure 2B). Notably, LPM239 achieved the highest suppression rates against both *S.* Typhimurium SL1344 (90.48%) and *S. marcescens* ATCC8100 (87.27%, Figure 2C).

The effects of LSM231, LCM233, and LPM239 on the morphology of pathogenic bacteria were examined via SEM. In the PBS control group, ETEC K88, *S. aureus* CMCC26003, *S.* Typhimurium SL1344, and *S. marcescens* ATCC8100 exhibited intact cellular morphology with regular shapes and smooth membrane surfaces. Following exposure to lactic acid bacteria metabolites, all four pathogen bacteria developed varying degrees of surface irregularities, including protrusions, ruptures, and depressions. These morphological alterations indicate that LSM231, LCM233, and LPM239 significantly compromised the structural integrity of the pathogen bacteria (Figure 3A). TEM was employed to evaluate lactic-acid-bacteria-induced damage to cellular membranes and cytoplasmic organization. PBS-treated pathogens maintained intact cell walls and membranes with homogeneous surface ultrastructure. After treatment with lactic acid bacteria metabolites, the pathogens exhibited the following: vacuolization within the cytoplasm, disintegration of the subcellular architecture with diffuse regions, and extensive damage to cell walls and membranes, including partial detachment and cytoplasmic leakage. These ultrastructural disruptions demonstrate potent membranolytic activity of the *Lactobacillus* strains against all tested pathogens (Figure 3B).

### 3.3. Synergistic Antimicrobial and Antiviral Activities of Multi-Strain Probiotic

Co-cultivation of the three lactic acid bacteria strains exhibited a significant synergistic growth effect. OD reached a stationary phase at 14 h and attained its maximum value at 24 h (Figure 4A). The probiotic consortium significantly enhanced the growth of the lactic acid bacteria (*p* < 0.001), increasing the relative growth rates by 7.5%, 12.14%, and 9.5% for strains LSM231, LPM239, and LCM233, respectively, compared to their individual monocultures (Figure 4B). The inhibitory zone sizes of the metabolites from the multi-strain probiotic against *S. aureus* CMCC26003, *S.* Typhimurium SL1344, *S. marcescens* ATCC8100, and ETEC K88 are presented in Figure 4C, with quantitative measurements detailed in Table 3. The cell-free supernatant of the multi-strain probiotic demonstrated notable inhibitory effects against all four tested pathogenic bacteria. Furthermore, the multi-strain probiotic significantly enhanced the inhibition rates against *S. aureus* CMCC26003 and *S. marcescens* ATCC8100 compared to the controls (*p* < 0.05). Both the supernatant and bacterial suspension of the multi-strain probiotic inhibited the replication of PEDV in IPI-2I cells, significantly reducing viral copy numbers (*p* < 0.001). Furthermore, compared to individual strains, the multi-strain probiotic elicited a significantly greater reduction in viral copy numbers (*p* < 0.05), thus providing preliminary evidence for its antiviral activity (Figure 4D,E).

### 3.4. Multi-Strain Probiotic Provided Protection to Piglets Against PEDV Infection

The piglets in the PEDV-infected group exhibited watery diarrhea, anorexia, a reduction in body weight, and progressive difficulty in feeding and drinking. In contrast, the piglets receiving the multi-strain probiotic showed only mild diarrheal symptoms while maintaining normal feeding and drinking behavior. Necropsy revealed that the PEDV group displayed intestinal wall thinning and translucency, gaseous distension of the intestines, and grossly visible yellow, watery intestinal contents. The multi-strain probiotic + PEDV group presented with slightly thinner intestinal walls and contained small amounts of grossly visible yellow, watery intestinal contents (Figure 5A). In the PEDV group, the pigs exhibited diarrhea 14 h post-infection, alongside symptoms of vomiting and diminished feed consumption, resulting in a daily reduction in body weight. Conversely, the multi-strain probiotic + PEDV group began to exhibit diarrhea 24 h following PEDV infection, with a reduced severity compared to the PEDV group. Although their daily weight gain surpassed that of the PEDV group, it did not reach the levels observed in the CON group (Figure 5B). No virus was detected in either the anal or nasal swabs from the CON group. In contrast, viral copy numbers in both the anal and nasal swabs of the piglets in the PEDV group increased significantly at 6 h post-infection (*p* < 0.001), and the virus remained detectable throughout the experimental period (Figure 5C). Conversely, the multi-strain probiotic + PEDV group exhibited a significant decrease in viral copy numbers in both the anal and nasal swabs (*p* < 0.05). Following PEDV infection, viral copy numbers in the jejunum and ileum of the piglets increased significantly (*p* < 0.001). Compared with the PEDV-infected group, the multi-strain probiotic + PEDV group exhibited a highly significant reduction in viral copy numbers in both the jejunum and ileum (*p* < 0.001), as illustrated in Figure 5D. The quantification of inflammatory cytokines (*IL-8*, *IL-18*, and *TNF-α*) in the ileal tissues revealed significantly higher mRNA expression levels in the PEDV group compared to the multi-strain probiotic + PEDV group (*p* < 0.05) (Figure 6A). The analysis of tight junction components demonstrated that the multi-strain probiotic significantly upregulated transcriptional levels of *ZO-1*, *Occludin*, and *E-cadherin* mRNA relative to the PEDV-infected piglets (*p* < 0.05) (Figure 6B). Furthermore, oral administration of the multi-strain probiotic markedly enhanced mRNA expression of interferon-stimulated genes (*ISG15*, *OASL*) versus the PEDV group (*p* < 0.01) (Figure 6C). These findings indicate that the multi-strain probiotic attenuates viral-induced intestinal barrier damage through dual mechanisms: the suppression of pro-inflammatory cytokine production and coordinated upregulation of both epithelial junctional proteins and antiviral ISGs.

### 3.5. Multi-Strain Probiotic Protected from Intestinal Damage During PEDV Infection

In the CON group piglets, the intestinal villi maintained intact architecture with no observable inflammatory cell infiltration. In contrast, the PEDV group exhibited varying degrees of intestinal epithelial damage, characterized by villus shortening, necrosis, lysis and desquamation of epithelial cells, abundant cellular debris within the intestinal lumen, and marked inflammatory cell infiltration in the lamina propria. Notably, the PEDV group demonstrated the most severe disruption of the intestinal epithelium. Following PEDV infection in piglets, villus height in the jejunum and ileum decreased, and the villus-to-crypt ratio was significantly reduced (*p* < 0.01). Compared with the PEDV-infected group, the multi-strain probiotic + PEDV group exhibited elevated villus height in both the jejunum and the ileum and an elevated villus-to-crypt ratio (*p* < 0.05). These histological improvements indicate that the multi-strain probiotic enhances intestinal nutrient absorption efficiency by increasing villus length and thereby expanding the absorptive surface area, while simultaneously fortifying the intestinal barrier function and reducing the translocation of harmful substances (Figure 7A,B). Following PEDV infection, the piglets exhibited a significant reduction in both goblet cell numbers and mucus layer thickness in the jejunum and ileum (*p* < 0.001). Compared with the PEDV group, the composite probiotics group had significantly increased goblet cell numbers and mucus layer thickness in the jejunum and ileum (*p* < 0.001) (Figure 7C,D).

SEM revealed more orderly and densely arranged intestinal villi with an intact structure in the probiotic consortium group compared to the PEDV-infected group (Figure 8A). The TEM of jejunum and ileum showed microvilli fragmentation and sparse arrangement in the PEDV-infected piglets, with indistinct tight junctions and widened intercellular spaces. In contrast, the multi-strain probiotic + PEDV group exhibited densely aligned microvilli with minimal fragmentation and clearly defined, continuous tight junctions (Figure 8B). These results collectively demonstrate that the multi-strain probiotic significantly mitigates PEDV-induced damage to intestinal microvilli and epithelial tight junctions.

## 4. Discussion

In pig farming, high levels of feed consumption, antibiotic use, and waste emissions are common [19]. Lactic acid bacteria offer a natural alternative to antibiotics, making their study important for breeding [20,21]. Min pigs, a valuable breed in Northeast China, possess traits like firm meat, high fertility, and cold resistance [22,23]. Identifying probiotics from their intestines is crucial for breed development. Specifically, *LimosiLactobacillus reuteri* and *Lactobacillus amylovorus* from Min pigs can resist PEDV infection [18]. Probiotics must endure the challenging environment of the gastrointestinal tract, reach the intestines in sufficient quantities, and colonize the intestinal mucosa to be effective [24,25]. Pepsin and trypsin can inhibit these bacteria by degrading their cell surface proteins [26]. This study found that three strains from Min pigs can resist low pH and bile salts, surviving at pH 3 and in 0.1% pig bile salts. After 3 h in artificial gastrointestinal fluid, they maintain over 40% survival, showing tolerance to digestive enzymes.

Assessing probiotics often involves considering their antimicrobial activity as a significant criterion [27]. Studies indicate that lactic acid bacteria can inhibit pathogens by producing antibacterial substances [28,29]. This study evaluated the antibacterial effects of three lactic acid bacteria strains from Min pigs against *S. aureus* CMCC26003, *S.* Typhimurium SL1344, *S. marcescens* ATCC8100, and ETEC K88. The results showed that LCM233, LSM231, and LPM239 inhibited all four pathogens, with LSM231 being most effective against *S. aureus* CMCC26003, LCM233 against ETEC K88, and LPM239 against *S. marcescens* ATCC8100 and *S.* Typhimurium SL1344, demonstrating strong antibacterial properties. Probiotics that include multiple strains are widely believed to be effective in promoting gut microbiota stability and host health, offering more advantages compared to single-strain probiotics [30,31]. This study combined three lactic acid bacteria strains into a compound probiotic. Using the Oxford cup method, it was found that this compound significantly enhanced antibacterial effects against *S. aureus* CMCC26003, *S.* Typhimurium SL1344, *S. marcescens* ATCC8100, and ETEC K88. Additionally, it significantly reduced PEDV virus replication in IPI-2I cells (*p* < 0.05).

PEDV continues to impact the global pig industry, with highly pathogenic G2-type strains (G2a, G2b) emerging since 2010, leading to high mortality rates in piglets [32]. The prevalence of G2b-type recombinant strains has increased, reducing the effectiveness of the current vaccines due to changes in the S protein antigenic epitope [33]. Probiotics confer health benefits to the host when administered in appropriate dosages, primarily by modulating the production of pro-inflammatory and anti-inflammatory cytokines by intestinal immune cells, thereby contributing to the maintenance of intestinal microbiota homeostasis [34]. Compound probiotics have demonstrated a significant capacity to regulate inflammatory factors in piglets infected with PEDV. Research indicates that PEDV infection induces excessive secretion of pro-inflammatory cytokines, such as TNF-α, IL-1β, and IL-6, by intestinal epithelial cells, which leads to intestinal inflammatory responses [35]. Compound probiotics mitigate the inflammatory response by modulating the MyD88/NF-κB signaling pathway [36]. This mechanism is intricately linked to the regulation of histone deacetylase by probiotic-derived short-chain fatty acids (SCFAs) [37]. The synergistic effect of a specific combination of probiotic strains is notably pronounced. A compound formulation containing *Bifidobacterium longum* has been shown to significantly reduce the mRNA expression levels of TNF-α in the colonic tissue of piglets infected with PEDV, outperforming single-strain interventions [38]. This regulatory effect is likely attributable to the restructuring of the intestinal microbiota by the compound probiotics, which increases the abundance of Bacteroidetes and decreases the proportion of Proteobacteria, thereby indirectly inhibiting the excessive activation of inflammatory signaling pathways [38]. To investigate the impact of compound probiotics on intestinal inflammatory markers in PEDV-infected piglets, this study examined the changes in mRNA levels of inflammatory factors in the jejunum and ileum. The results indicated that PEDV infection significantly upregulated the expression levels of pro-inflammatory factors IL-8, TNF-α, and IL-18 (*p* < 0.01). However, in the group receiving compound probiotic treatment, the expression levels of IL-8, TNF-α, and IL-18 were significantly reduced (*p* < 0.05). These results suggest that compound probiotics have the potential to mitigate the expression of inflammatory factors and alleviate the inflammatory damage induced by PEDV.

The development and health of the intestines are significantly marked by villus height and crypt depth [39]. A recent study has demonstrated that probiotics can markedly increase the intestinal villus height/crypt depth ratio of piglets infected with PEDV, thereby facilitating the regeneration and repair of intestinal epithelial cells [11]. Moreover, the augmentation of compound probiotics, including *Lactobacillus rhamnosus* and *Bifidobacterium*, has been shown to diminish the colonization of pathogenic bacteria, thereby preserving intestinal health [40,41]. This experiment investigated the alterations in the intestinal morphology of the jejunum and ileum in PEDV-infected piglets. The findings revealed a significant reduction in intestinal villus height/crypt depth ratio following PEDV infection (*p* < 0.01). Conversely, in the group receiving compound probiotic protection, there was a significant increase in intestinal villus height/crypt depth ratio (*p* < 0.05). Additionally, observations using scanning electron microscopy and transmission electron microscopy corroborated that compound probiotics substantially mitigate intestinal damage.

As the primary site for digestion and absorption, the intestine is continuously exposed to a multitude of exogenous substances, making the integrity of its barrier function critically important [42,43]. This integrity is pivotal in determining the intestine’s ability to effectively resist and eliminate toxins and pathogens, thereby maintaining the proper functioning of intestinal defense mechanisms [44]. PEDV infection can elevate the incidence of diarrhea in piglets, trigger a pro-inflammatory response, and compromise the intestinal barrier [45]. Probiotics have been shown to ameliorate the intestinal barrier by modulating the expression of tight junction proteins and enhancing mucus layer secretion [46]. In this study, the mRNA expression levels of tight junction proteins, including Claudin-1, Occludin, and E-cadherin, were assessed in the jejunum and ileum of PEDV-infected piglets. The results indicated that the group receiving compound probiotic protection exhibited a significant upregulation in the expression levels of these tight junction proteins (*p* < 0.05). Furthermore, the protein expression level of Occludin was assessed, and the findings were consistent with the mRNA expression levels. Concurrently, AB-PAS staining of the intestinal mucus layer revealed that the thickness of the mucus layer and the density of goblet cells in the compound probiotic protection group were significantly greater than those observed in the PEDV infection group (*p* < 0.001). This evidence further corroborates the hypothesis that compound probiotics can preserve the integrity of the intestinal mucosa by enhancing the expression of intestinal tight junction proteins and increasing the thickness of the mucus layer, thereby providing resistance against PEDV infection.

Viral invasion can elicit an immune response, subsequently leading to alterations in the expression of antiviral proteins [47,48]. Studies demonstrate that probiotics can significantly enhance the levels of IFN-α and ISGs in the intestines of piglets suffering from PEDV infection [49,50]. This upregulation occurs through the activation of the host’s innate immune response, which enhances the levels of antiviral proteins and thereby fortifies the piglets’ defense against PEDV [49]. Furthermore, research conducted by Weiss et al. demonstrated that probiotics augment the expression of antiviral proteins by modulating the Toll-like receptor signaling pathway, thereby improving the immune response capability of piglets to PEDV [50]. In this in vivo study, the modulation of the expression level of the antiviral protein ISG15 by compound probiotics was examined. Following oral administration of compound probiotics to the piglets, both the mRNA transcription level and the protein expression level of ISG15 were elevated compared to those in the PEDV-infected control group.

## 5. Conclusions

The strains *L. crispatus* LCM233, *L. salivarius* LSM231, and *L. plantarum* LPM239 exhibit notable antimicrobial activity and stress tolerance. When formulated as a multi-strain probiotic, these lactic acid bacteria demonstrate significant synergistic growth, resulting in enhanced antibacterial and antiviral efficacy. In piglets infected with PEDV, this composite probiotic supports the maintenance of intestinal barrier homeostasis by repairing villi and tight junctions, increasing the number of goblet cells, and reducing the expression of inflammatory cytokines to mitigate virus-induced damage. Additionally, it enhances the expression of barrier proteins to preserve mucosal integrity and upregulates antiviral proteins to combat infection. This study elucidates the protective effects of a composite probiotic on piglets, presenting a novel strategy for the prevention of PED and for the development of microecological live bacterial formulations to alleviate intestinal damage in piglets resulting from PEDV infection.

## Figures and Tables

**Figure 1 microorganisms-13-01738-f001:**
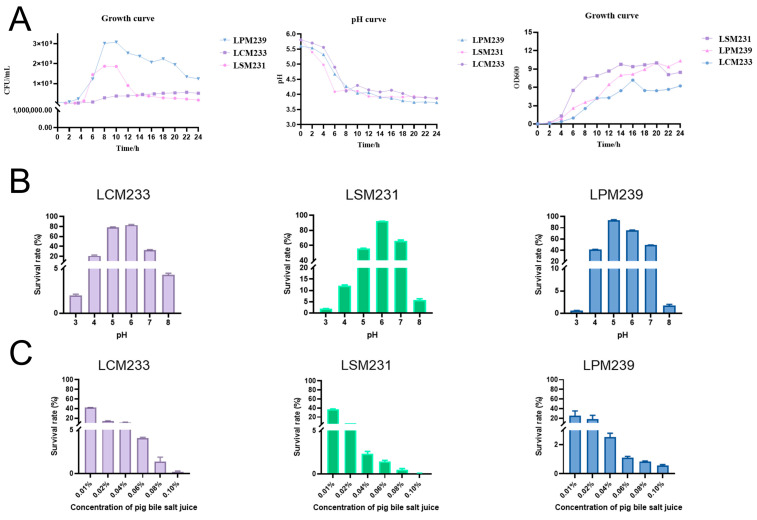
Probiotic traits of Min pig gut-derived lactic acid bacterial strains LCM233, LSM231, and LPM239. (**A**) Growth kinetics and acid production profiles of LCM233, LSM231, and LPM239. (**B**) Acid-base tolerance profiling of LCM233, LSM231, and LPM239. (**C**) Differential bile salt tolerance of LCM233, LSM231, and LPM239.

**Figure 2 microorganisms-13-01738-f002:**
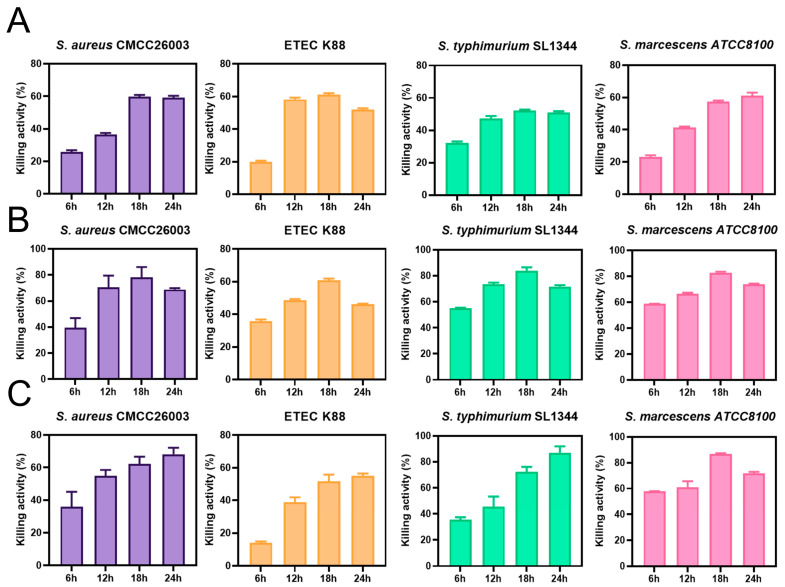
In vitro antimicrobial activity of LCM233, LSM231, and LPM239. Growth inhibition rates of LCM233 (**A**), LSM231 (**B**), and LPM239 (**C**) against pathogenic bacteria.

**Figure 3 microorganisms-13-01738-f003:**
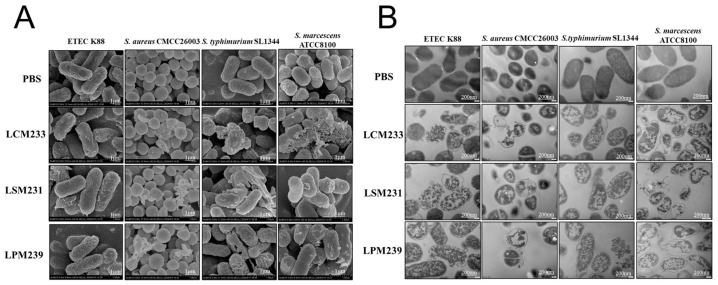
Morphopathological alterations in pathogens. (**A**) SEM analysis of pathogen morphological disruption induced by *Lactobacillus* strains (35,000×). (**B**) TEM visualization of pathogen membrane ultrastructure alterations induced by *Lactobacillus* strains (50,000×).

**Figure 4 microorganisms-13-01738-f004:**
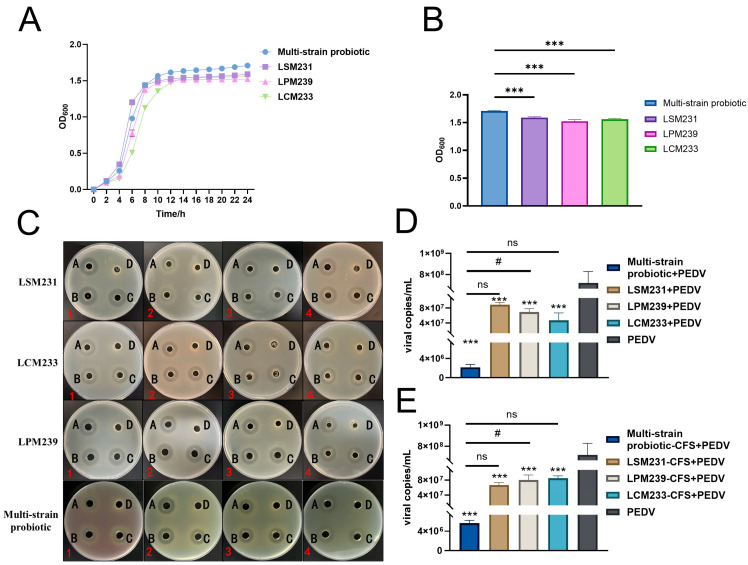
The multi-strain probiotic exhibited superior antibacterial and antiviral activity. (**A**) Growth curve. (**B**) The 24 h growth dynamics of multi-strain probiotic. (**C**) Inhibition zones of LCM233, LSM231, LPM239, and multi-strain probiotic. A, B, C: Bacterial culture supernatant; D: Control; 1: *S.* Typhimurium SL1344; 2: ETEC K88; 3: *S. aureus* CMCC26003; 4: *S. marcescens* ATCC8100. (**D**) Effect of bacterial suspension on PEDV-infected IPI-2I cells. (**E**) Effect of bacterial culture supernatant on PEDV-infected IPI-2I cells. ns indicates not significant; *** *p* < 0.001 vs. PEDV, ^#^ *p* < 0.05 vs. Multi-strain probiotic-CFS+PEDV.

**Figure 5 microorganisms-13-01738-f005:**
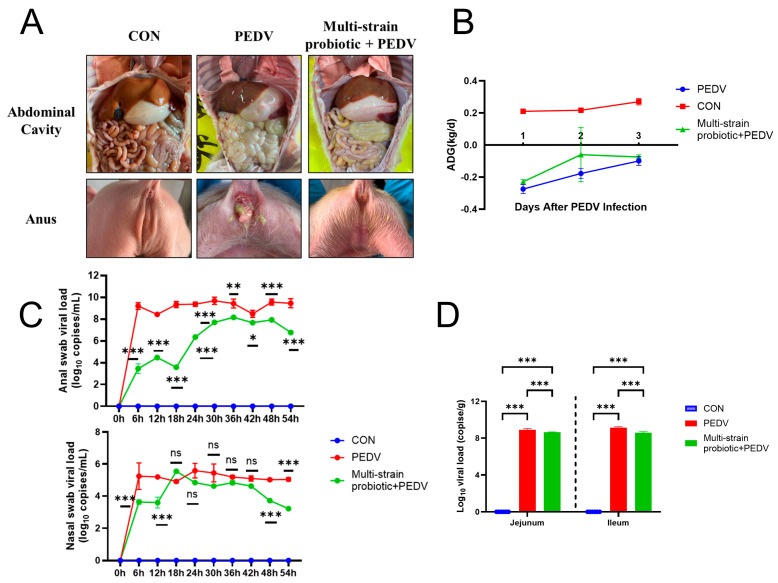
The antiviral effects of multi-strain probiotic during PEDV infection. (**A**) Clinical symptoms and intestinal pathological alterations. (**B**) Changes in average daily gain. (**C**) Viral load in piglet anal and nasal swabs. (**D**) Viral load in piglet jejunum and ileum. ns indicates not significant; * *p* < 0.05, ** *p* < 0.01, *** *p* < 0.001.

**Figure 6 microorganisms-13-01738-f006:**
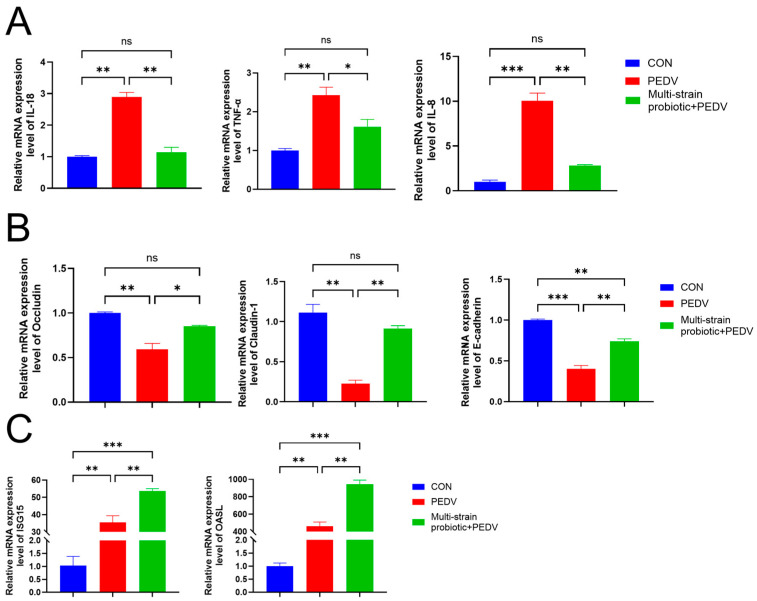
Host response gene expression in PEDV-infected piglets. (**A**) mRNA expression levels of inflammation-related genes (*IL-18*, *TNF-α*, and *IL-8*). (**B**) mRNA expression levels of intestinal-barrier-associated genes (*Occludin*, *Claduin-1*, and *E-cadherin*). (**C**) mRNA expression of interferon-stimulated genes (*ISG15* and *OASL*). ns indicates not significant; * *p* < 0.05, ** *p* < 0.01, *** *p* < 0.001.

**Figure 7 microorganisms-13-01738-f007:**
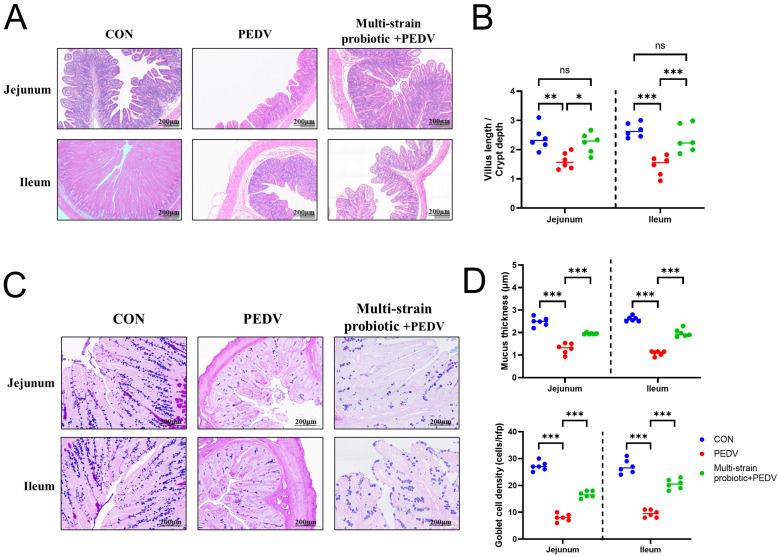
Multi-strain probiotic alleviates intestinal barrier damage in PEDV-infected piglets. (**A**) Pathological changes in the small intestine of piglets. (**B**) Measurement of villus height/crypt depth in the small intestine of piglets. (**C**) AB/PAS staining for mucin detection in piglet ileal tissue. (**D**) Changes in goblet cell numbers and mucus layer thickness in the jejunum and ileum of piglets. ns indicates not significant; * *p* < 0.05, ** *p* < 0.01, *** *p* < 0.001.

**Figure 8 microorganisms-13-01738-f008:**
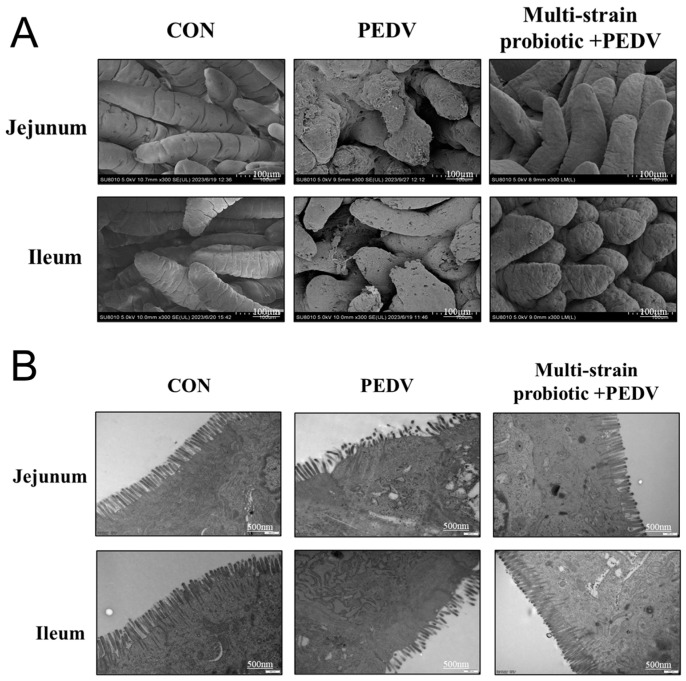
Effect of multi-strain probiotic on the ultrastructure of intestinal villi in piglets. (**A**) SEM results of jejunal and ileal tissue in piglets. (**B**) TEM analysis of ultrastructure.

**Table 1 microorganisms-13-01738-t001:** Primer sequences.

Targets	Primer (5′-3′)	Accession Number
*PEDV-N*	F: 5′-GGTATTGGAGAAAATCCTGACAGGCATAAGCAACAGCA-3′	Gene sequencing was accomplished by our laboratory without reference to databases.
	R: 5′-GACGCATCAACACCTTTTTCGACAAATTCCGCATC-3′	
*β-actin*	F: 5′-CGGGACATCAAGGAGAAGC-3′	NC_010445.4
	R: 5′-ACAGCACCGTGTTGGCGTAGAG-3′	
*IL-8*	F: 5′-CCACACCTTTCCACCCCAAA-3′	NM_213867.1
	R: 5′-GGCTCCGGTTCGACACTTTC-3′	
*TNF-α*	F: 5′-CACCACGCTCTTCTGCCTAC-3′	X57321.1
	R: 5′-GGCTTTGACATTGGCTACAACG-3′	
*IL-18*	F: 5′-TCTACTCTCTCCTGTAAGAAC-3′	NM_213997.1
	R: 5′-CTTATCATGTCCAGGAAC-3′	
*IL-6*	F: 5′-GCCTTCAGTCCAGTCGCCTTCT-3′	NM_214399.1
	R: 5′-GTGGCATCACCTTTGGCATCTTC-3′	
*E-cadherin*	F: 5′-CCCCAACACTTCTCCCTTCACT-3′	NM_001163060.1
	R: 5′-CTCGAGGGTTTTCTTTGGCTTC-3′	
*ZO-1*	F: 5′-CTCTTGGCTTGCTATTCG-3′	XM_003353439.2
	R: 5′-AGTCTTCCCTGCTCTTGC-3′	
*Occludin*	F: 5′-GTAGTCGGGTTCGTTTCC-3′	NM_001163647.2
	R: 5′-GACCTGATTGCCTAGAGTGT-3′	
*Integrin*	F: 5′-GCAGTTTCAAGGTCAAGATGG-3′	NM_214002.1
	R: 5′-AGCAGGAGGAAGATGAGCAG-3′	
*Claudin-1*	F: 5′-AGATTTACTCCTACGCTGGTGAC-3′	NM_001244539.1
	R: 5′-GCAAAGTGGTGTTCAGATTCAG-3′	
*OASL*	F: 5′-GGCACCCCTGTTTTCCTCT-3′	NM_001031790.1
	R: 5′-AGCACCGCTTTTGGATGG-3′	
*ISG15*	F: 5′-AGCATGGTCCTGTTGATGGTG-3′	NM_001128469.3
	R: 5′-CAGAAATGGTCAGCTTGCACG-3′	

**Table 2 microorganisms-13-01738-t002:** Viability of LCM233, LSM231, and LPM239 in simulated gastrointestinal fluids.

Strains	Survival Rate (%)	
	Simulated Gastric Fluid	Simulated Intestinal Fluid
LCM233	81.82 ± 12.86 ^a^	48.64 ± 5.36 ^a^
LSM231	64.06 ± 4.52 ^b^	58.4 ± 7.92 ^b^
LPM239	80.00 ± 4.71 ^a^	42.19 ± 8.14 ^c^

Values are expressed as mean ± SD. ^a–c^: Means not sharing the same letter superscript differ significantly (*p* < 0.05).

**Table 3 microorganisms-13-01738-t003:** Diameter of inhibition zones in multi-strain probiotic assays in vitro.

Strains	Inhibition Zone Diameters (mm)			
	*S. aureus* CMCC26003	ETEC K88	*S. marcescens* ATCC8100	*S.* Typhimurium SL1344
LCM233	17.27 ± 0.17 ^a^	17.35 ± 0.22 ^a^	16.64 ± 0.33 ^a^	16.67 ± 0.2 ^a^
LSM231	18.25 ± 0.05 ^b^	18.93 ± 0.11 ^b^	18.06 ± 0.5 ^b^	19 ± 0.51 ^b^
LPM239	18.16 ± 0.49 ^b^	18.64 ± 0.41 ^b^	17.84 ± 0.17 ^c^	17.48 ± 0.26 ^c^
Multi-strain probiotic (1:1:1)	19.42 ± 0.24 ^c^	19.47 ± 0.21 ^c^	18.86 ± 0.29 ^d^	19.01 ± 0.47 ^d^

Values are expressed as mean ± SD. ^a–d^: Means not sharing the same superscript letter differ significantly.

## Data Availability

The original contributions presented in this study are included in the article. Further inquiries can be directed at the corresponding authors.

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
