# Peer review of "Multi-Strain Probiotic Regulates the Intestinal Mucosal Immunity and Enhances the Protection of Piglets Against Porcine Epidemic Diarrhea Virus Challenge"

_microorganisms, 2025, doi:10.3390/microorganisms13081738_

Round 1
Reviewer 1 Report
Comments and Suggestions for Authors
This is a very well structured study, applying a wide range of techniques to come to an important endpoint, the production of a novel probiotic soup. The authors carried out a reasonable rationale to evaluate its utility and performance against various conditions. I am totally positive for acceptance of this work, but I have a few points that are not well clarified and need clarification. You can find these points below:
Is PEDV always pathogenic or there are also asymptomatic pigs? Is it also zoonotic, i.e. could it be transmitted to other animals and human? Please add this info to enhance the importance and validity of this work
The description of the selection of the three Lactobacillus strains for the probiotic is sufficient, but it is not well understood why in Section 2.1 these bacteria were isolated. Please explain the rationale, they are not related with the three Lactobacillus probiotics
The PEDV group challenge; has to be more in detail explained. How were they infected?
2.11.1 Only from the tissue lesions or randomly?
In Table 1, please provide a reference for each primer pair or the design procedure
Reviewer 2 Report
Comments and Suggestions for Authors
It´s a beautiful and very important research.
New nomenclature for the Lactobacillus group should be used. For example, Lactiplantibacillus plantarum. In some parts of the text is written Lactobacillus salivarius and in other parts LigiLactobacillus salivarius.
Why were those probiotic strains selected? Did these strains have any special characteristics? Maybe a table showing their characteristics.
2.3. Validation of Synergistic Effects in Multi-strain Probiotic: It is not clear what synergy is intended to be measured at this point. Maybe a scheme should be added to explain this experiment.
The fluid contents composition (gastric and intestinal) should be described.
Salmonella italicized, but not Typhimurium (not italicized, capitalized), because Typhimurium is the serovar.
Bifidobacterium longum (italics)
